# Mediator Effect of Affinity for E-Learning on Mental Health: Buffering Strategy for the Resilience of University Students

**DOI:** 10.3390/ijerph18137098

**Published:** 2021-07-02

**Authors:** Dina Di Giacomo, Alessandra Martelli, Federica Guerra, Federica Cielo, Jessica Ranieri

**Affiliations:** 1Department of Life, Health and Environmental Sciences, University of L’Aquila, 67100 L’Aquila, Italy; federica.guerra@graduate.univaq.it (F.G.); federica.cielo@graduate.univaq.it (F.C.); jessica.ranieri@univaq.it (J.R.); 2Postgraduate School on Clinical Psychology, University of L’Aquila, 67100 L’Aquila, Italy; 3Faculty of Biosciences and Agri-Food and Environmental Technologies, University of Teramo, 64100 Teramo, Italy; alessandra.martelli@unite.it

**Keywords:** COVID-19, pandemic, psychological impact, mental health, young

## Abstract

The pandemic affected the quality of life and wellness of the population, changing living habits through restriction measures. This study aimed to analyze the psychological impact of the fear of the COVID-19 pandemic and the adoption of e-learning for university students. The study was articulated in two research applications: the first application was a rapid review on the psychological effects of the pandemic on the emotional dimension of undergraduate students; the second application was an observational study on the effect of e-learning adoption in the pandemic emergency. In the first step, we performed a systematic search of MEDLINE through PubMed and the Web of Science [Science Citation Index Expanded (SCI-EXPANDED); Social Sciences Citation Index (SSCI); Emerging Sources Citation Index (ESCI)] of all scientific literature published from May 2020 to February 2021. The reviewed articles suggest the impact of the pandemic and lockdown measures on university students due to several mental symptoms, including anxiety, stress, depression, event-specific distress, and a decrease in psychological well-being. Psychological symptoms were related to the experience of several stressors, such as the risk for a reduction of academic perspectives, massive e-learning adoption, economic issues, social restrictions, and implications for daily life related to the COVID-19 outbreak. The second scientific application was conducted to evaluate the affinity for e-learning on a sample composed of Italian undergraduates exposed to massive e-learning adoption. The results evidence the positive influence of e-learning in academic programs for the wellbeing of undergraduates. The mediator effect of the affinity of youth for e-learning can be considered to have had a buffering effect for professional advancement and for the mental health of university students in a public health emergency.

## 1. Introduction

In March 2020, the World Health Organization (WHO) declared a state of pandemic [1]. The clinical features of COVID-19 are varied, ranging from an asymptomatic state to acute respiratory distress syndrome and multi-organ dysfunction. COVID-19 provoked significant challenges to curb the spread of the infection and maintain global health security [2,3,4]. Due to the rapid spread of the coronavirus, many countries implemented a range of anti-epidemic measures, such as keeping a physical distance, wearing a face mask, quarantine, and lockdown restrictions to contain transmission and avoid contact with others. Furthermore, the COVID-19 outbreak affected not only physical health, but also mental health and wellbeing [5,6,7]. The COVID-19 pandemic has severely affected the entire general population. Even young people have not been spared from the changes of this unprecedented situation [8]. Psychosocial outcomes on young university students have been detected at the early stages as well at the prolonged stage of the pandemic, evidencing mental health outcomes. Particularly, the protective measure adopted in academic programs was the massive adoption of e-learning to preserve the educational needs of youth even in lockdown restrictions.

Several studies analyzed the impact of the pandemic (in regard to social isolation and restrictive measures) on the mental health of the young generation, and more specifically, undergraduates. No research investigated the factorial aspects and consequences of the sudden changing of educational strategy based fully on e-learning methodology. Academic programs are very different through e-learning; teaching as well laboratory activities became fully online.

The present study aimed to investigate the psycho-behavioral impact of COVID-19 on the quality of life of academic students. The study was articulated in two study protocols: the first scientific application was aimed at reviewing the emerging literature on COVID-19 on mental health outcomes among youth. Further, we wanted to analyze COVID-19-related risk factors to capture potentially stressful events related to the increase in the spread of the coronavirus. The second scientific application was an observational study focused on the affinity for e-learning of university students after massive e-learning adoption, and we wanted to verify if e-learning could be a protective factor for youth conducting mediation analyses. We wanted to verify the impact of e-learning as a strategy for the educational perspectives of undergraduates. Our scope was to examine their fears about their professional training and failure of academic performance in regard to the fields of academic programs (social and human sciences, life sciences, and physical and engineer sciences).

## 2. First Study Protocol: Rapid COVID Review on Academic Students

### 2.1. Aim of the Rapid Review

We conducted a qualitative analysis to map the literature on the psychological impact of the COVID-19 outbreak among youth. Our aim was to synthesize current available scientific investigations related to mental health aspects and to illustrate the educational scenario.

### 2.2. Search Strategy

To identify potentially relevant studies for inclusion, we performed a systematic search of MEDLINE through PubMed and the Web of Science [Science Citation Index Expanded (SCI-EXPANDED); Social Sciences Citation Index (SSCI); Emerging Sources Citation Index (ESCI)] in the time range of March 2020–February 2021, with the terms ‘COVID-19′, ‘psychological impact’, ‘mental health disorders’, ‘lockdown’ and ‘undergraduate’.

### 2.3. Inclusion and Exclusion Criteria

We included all papers related to the psychological impact of the COVID-19 outbreak published in English until February 2021 that are related to university students. However, we excluded reports that were not published in scientific peer-reviewed journals, reviews, and studies in which the sample target (age range of 18–30 years old) was not young people.

### 2.4. Article Selection and Data Extraction

To ensure the reliability of the narrative review, two reviewers independently screened all publication titles and abstracts for inclusion. The following information was extracted from each paper: (a) authors, (b) title, (c) source title, (d) publication year, (e) age range, and (f) topic (COVID-19, coronavirus, mental health).

### 2.5. Statistical Analysis

We conducted a descriptive analysis of the characteristics of the included literature to examine the psychological impact of the COVID-19 outbreak among young people. We conducted this review in accordance with the PRISMA flow diagram [9].

### 2.6. Search Results

The literature search of the PubMed and Web of Science databases provided a total of 199 publications. After removing duplicates, 179 papers were identified for screening. Based on the criteria, two reviewers screened all publication titles and abstracts for inclusion, and successfully resolved all disagreements by consensus.

Figure 1 illustrates the study selection flowchart.

Of these, we excluded records that were clearly off-topic papers, reviews, articles only focused on psychiatric patients, patients with medical co-morbidity, pregnant women, community members, children, and/or adults. The full text of the remaining 28 citations was examined comprehensively and identified as potentially eligible articles. A total of 15 studies did not meet the inclusion criteria. Finally, 13 articles were included in the present rapid review.

### 2.7. Characteristics of Included Articles

Table 1 synthesizes the main characteristics of the included studies. We extracted the following from each included paper: (a) study design, (b) authors, (c) sample size, (d) sample, and (e) recruitment. All the papers were grouped according to the study design (cross-sectional study and longitudinal study) and include undergraduate participants. Furthermore, due to the pandemic situation, lockdown restrictions, and extraordinary measures to curb the spread of the coronavirus, almost all of the studies collected data through online surveys.

Regarding the research design, the articles included in the present rapid review were 12 cross-sectional studies [10,11,12,13,14,15,16,17,18,19,20,21] and 1 longitudinal study [22]. Generally, the survey link was disseminated in various platforms available on social media. Only one study [14] did not mention the recruitment method. All studies were conducted in the time range of March–June 2020.

### 2.8. Type of Articles

Table 2 highlights the details of the screened articles that were grouped into cross-sectional and longitudinal studies. We extracted the following from each included paper: (a) study design, (b) topic, (c) measures, and (d) outcomes. All the papers focus on the impact of COVID-19 on mental health, while outcome tools mainly measured an increased risk of psychological disorders, such as anxiety, depression, general distress, and event-specific distress.

### 2.9. Overview of the Psychological Effects of COVID-19 for University Students

The outcomes of the included studies showed that youth experienced anxiety [10,11,12,13,15,16,17,18,19,20,21,22], stress [17,18,21], depression [15,17,18,19,21], and event-specific distress [17] during the COVID-19 outbreak. One study [14] underlined a decrease in perceived psychological wellbeing because of the lockdown. Moreover, higher than normal levels of somatization, obsessive–compulsive disorder, phobic anxiety, and paranoid ideation were found [16]. Other studies [10,11,12,13,14,15,16,17,18,19,20,21] explored how the COVID-19 pandemic influenced the prevalence of psychological symptoms. One [22] reported that the personality trait of tending to experience a high level of worrying could determine an anxiety response to the quarantine. High worriers before the lockdown showed a significant increase of anxiety and fear of mental health in comparison to low worriers during lockdown conditions. Young people with high worry were more anxious and had a lower sense of control [22]. The results of the included studies indicated that many factors were associated with psychological symptoms during the COVID-19 crisis. The female gender was associated with higher levels of anxiety during the COVID-19 pandemic compared to the male gender [10,11,19,20]. This result is in contrast to what is reported by another study indicating that males and females experienced similar levels of psychological symptoms as a result of the pandemic [12]. The research also underlines that those levels of anxiety were significantly different according to age. Younger individuals experienced more anxiety compared to older ones [19,20,21].

Moreover, along with the learning process, challenges of e-learning, delays of academic programs, and concerns for academic performance were found to be risk factors for anxiety symptoms [10,11,12,13,15,16,20]. Healthcare and medical students had a lower risk of developing anxiety compared with students in other fields of study [20].

The results of the included studies also revealed that young people experienced depression symptomatology during the COVID-19 outbreak [15,17,18,19,21]. Most of the research did not detect any gender difference; only one study highlighted female weakness related to high rates of emotional issues and depressive symptoms [19]. Finally, taking into account the age variable, the youngest students experienced higher levels of depression than the oldest [19,21].

Among the learning conditions, most of the young were getting depressed due to concern about their academic performance and the forced termination of their internships [15]. Regarding the field of study, students who engaged in health science-related studies had less risk of developing depression than students in other fields [18].

The primary stressors for undergraduates were having relatives or friends experience COVID-19 infection [11,12,18], financial uncertainty [12,15,17,20], the worsening of interpersonal conflict, and restrictions to social contact [19].

Even though the lockdown measures have been efficient for the decrease of COVID-19 infection, they favored several side effects. Regarding mental health and wellbeing, the most analyzed variables were anxiety, mental distress, and depression. Other analyzed variables included event-specific distress, quality of sleep, and psychological wellbeing. Examining the social restrictions to contain the spread of the coronavirus, most of the included studies collected data through online surveys, showing the positive impact of digital solutions. In general, the survey links were disseminated on various platforms available on social media (e.g., Facebook, WhatsApp, email, Google Forms).

Among the factors that were associated with psychological symptoms related to the COVID-19 outbreak, as expected, universities adopted massive e-learning. intensive stressors have been (a) the unforeseen changing of academic learning processes, and (b) the reducing relationship. So far, no study has been conducted to investigate if these changes could have affected the mental health of university students. Some research analyzed the impact of e-learning in academic program fields; medical students had a lower risk of developing anxiety and depression compared with students in other programs [18,20]. This might be because medical students might have been well-informed on the effects of the pandemic compared to other students. Emerging research topics could be investigating the changes in academic learning and psycho-behavioral dimensions and how these changes could be turned into a lesson learned for a public health emergency.

## 3. Second Study Protocol: Observational Study on Affinity for E-Learning of University Students

### 3.1. Aim of the Study

The aim of the study was to analyze the impact of the massive adoption of e-learning on university students in the COVID-19 emergency. The scope of the study was to investigate the positive and negative effects of digital settings of learning in academic programs. Particularly, we wanted to analyze the influence of massive e-learning adoption on the self-perception of undergraduates about the risk of an unsuccessful academic program, fear about their professional training, and failure of academic performance.

Informed consent was obtained from each participant, and this study adhered to the principles outlined in the Declaration of Helsinki.

### 3.2. Participants

The participants were 2021 Italian undergraduate students attending academic programs at the University of L’Aquila and the University of Teramo (Italy) (mean = 21.8, SD ± 1.97); the mean age of the sample is 23.2 years (SD = 4.7), and 79.6% of them were women. Most of the participants were pursuing a bachelor’s degree (59.5%). In addition, 40.9% of the participants were off-campus students, and there was no change in the percentage of students who were living with their families (87.3%) between the first (March 2020) and second (December 2020) COVID waves. The academic program fields of participants were as follows: human and social sciences (HSS) (n = 776, 38.4%); life sciences (LS) (n = 494, 24.4%); and physical and engineering sciences (PES) (n = 751, 37.2%). Participants were recruited from a dedicated university community using social media platforms. The inclusion criteria were as follows: (a) aged 18–30 years, (b) undergraduate students, and (c) provision of informed consent.

### 3.3. Measurements

The socio-demographic characteristics of the participants, namely, age, relationship status, university degree and course, student university condition, study area, and living arrangement during the first and second lockdowns in Italy, were assessed using a socio-demographic form. We also assessed anxiety symptoms, peritraumatic dissociation, stress, and affinity for e-learning. The psychological battery consisted of 5 self-report measures, which assessed anxiety and distress. These measures were used to ascertain the presence of psychological symptoms and assess their severity. An ad hoc questionnaire was used to assess e-learning affinity.

#### 3.3.1. Emotional Measures

Peritraumatic Dissociative Experiences Questionnaire (PDEQ) [23,24]. This 10-item self-report questionnaire measures the dissociation experienced during or immediately after a traumatic event. The PDEQ has well-established psychometric properties, and higher total scores indicate increased peritraumatic dissociation. A score of > 15 is indicative of significant dissociation. The Cronbach’s value was excellent (α = 0.997).

COVID-19 Student Stress Questionnaire (CSSQ) [25]. The CSSQ measures COVID-19-related sources of stress among undergraduate students. It consists of 7 items, which are rated on a 5-point scale that ranges from 0 (‘not at all stressful’) to 4 (‘extremely stressful’). The CSSQ consists of the following three subscales, which assess COVID-19-related stressors among students: (1) relationships and academic life (i.e., relationships with relatives, colleagues, professors, and academic activities), (2) isolation (i.e., social isolation and couples’ relationships, intimacy, and sexual life), and (3) fear of contagion. This scale also yields a global stress score, which can range from 0 to 28. The Cronbach’s value was good (α = 0.7).

Coronavirus Anxiety Scale (CAS) [26,27]. The CAS is a brief 5-item mental health screening tool that can be used to detect dysfunctional anxiety associated with the COVID-19 crisis. Each item assesses a unique manifestation of this particular form of anxiety. Specifically, it assesses the cognitive (i.e., rumination, worry, information processing biases, daydreaming, and planning), behavioral (i.e., dysfunctional activities, avoidance, and compulsive behaviors), emotional (i.e., fear, anxiety, and anger), and physiological (i.e., sleep disturbances, somatic distress, and tonic immobility) dimensions of coronavirus anxiety. Each item is rated on a 5-point scale that ranges from 0 (‘not at all) to 4 (‘nearly every day’), and these ratings indicate the frequency with which the symptom has been experienced during the past two weeks. A total score of ≥ 9 is indicative of dysfunctional anxiety. The Cronbach’s value was good (α = 0.8).

#### 3.3.2. Affinity for E-Learning Measure

Affinity for e-learning Questionnaire (AEQ). The AEQ is an experimental self-report measure that assesses affinity for e-learning. It is composed of 10 items, which are rated on a 5-point Likert scale that ranges from ‘completely disagree’ to ‘completely agree’. It assesses self-confidence in relation to service access, convenience and flexibility, lesson attendance, involvement, and information technology skills. A pilot study was conducted using a sample drawn from the target population (not included in this study) to examine the reliability of the AEQ. The Cronbach’s value was good (α = 0.8).

### 3.4. Procedure

Participants were recruited using snowball sampling. Participants were recruited from a dedicated university community using social media. This study was conducted during 1–28 February 2021. The self-report questionnaire was linked to the post, and participants could access it after providing written informed consent. The online self-report questionnaire lasted approximately 15 min. The data were stored in a dedicated server. Participation in the study was voluntary.

### 3.5. Statistical Analyses

Statistical analyses were performed with SPSS Statistics version 24 (IBM Corp., Armonk, NY, USA). ANOVA analyses and a post hoc test (Scheffe) were used to identify the relationships between the tested variables.

The mediation analysis was performed using Jamovi [28] to verify the effect of the ‘e-learning on stress’ variable. An α level of 0.05 was used in all presented analyses.

### 3.6. Results

#### 3.6.1. Descriptive Analyses of Psychological Impact on University Youth

We analyzed the performance of the participants in the psychological evaluations by academic program field. Table 3 summarizes the raw score data.

ANOVA analysis comparing emotional dimensions (PDEQ, CSSQ, and CAS) × academic program field (HSC, LF, PES) showed a significant high risk for dissociative responses (F(2,2018) = 3.03; η^2^ = 0.003; *p* = 0.04) by PES students (*p* = 0.01), whereas HSS seemed protected. No difference was detected between generalized anxiety disorder (CAS), coronavirus stress (CSSQ), and global stress and the three related indexes (relationship and academic life, isolation, fear of contagion). The academic program fields did not seem related to the stress from the pandemic. We processed the ANOVA analysis for affinity for e-learning and academic fields. The statistical analysis evidenced a significant effect (F(2,2018) = 9.07; η^2^ = 0.008; *p* = 0.001). In Table 4, the post hoc analysis is presented.

As reported in Table 4, HSS students seemed more confident with e-learning adoption during the pandemic. In contrast, LS and PES students appeared less flexible with this kind of teaching. In our opinion, the effects of less affinity in both fields are not related to digital skills, as all students are able to use technology for education. During the pandemic, LS and PES students have had to decrease (and, in some ways, give up) their practical learning. The specificity of LS and PES academic programs is characterized by intensive laboratory activities and clinical/experimental practice, and e-learning cannot substitute those parts of professional training.

#### 3.6.2. Mediation Analysis

A mediation model was used to examine whether affinity for e-learning mediated the stress of university students. In the hypothesized mediation model (see Figure 2), academic programs (LS, PES, HSS) were the independent variable, affinity e-learning (AEQ) was the mediator variable, and COVID-19 Student Stress (CSSQ) was the dependent variable. As shown in Figure 2, affinity e-learning mediated the stress of students (SE = 0.01, β = −0.44, t = −22.2, *p* = 0.001); however, there was no direct effect of the academic program.

## 4. Discussion

The COVID-19 experience was complex because it affected worldwide daily living. As summarized in our rapid review (the first study protocol), the early COVID-19-tailored studies detected the mental health implications for all targets (general population, healthcare workers, women, children, vulnerable, and fragile people); public health measures and social restrictions were adopted extensively.

A relevant target is represented by university students. Restrictive measures and social isolation-induced stress and mental health issues, significantly depending on the fear for their professional training. The rapid review evidenced the emotional traits affected in the early stages. One relevant stressor was the uncertain expectations for the future, and this challenge was dealt with by a massive adoption in academic programs of e-learning, continuing to ensure advanced teaching to youth. As highlighted in our first scientific application, the COVID-19 experience was stressful for undergraduates, as they were affected by anxiety, depression, changes in quality of sleep, and other mental symptoms [10,11,12,13,14,15,16,17,18,19,20,21,22]. A relevant common point of those early studies was the timing of psychological evaluations; most of them were conducted in the first semester of 2020 (January–May). The finding of literature tailored to COVID-19 early on led to the detection of the mental health impact of the pandemic in real-time, outlining the undergraduates’ stressful daily living, and providing a large database of the first semester of 2020.

In 2021, the pandemic was a prolonged experience, whereas the national vaccination plans progressively reduced the large risk for the spread of COVID-19; however, the effect on the mental health of the population still needs to be explored for the long-term outcomes. Our second scientific application focused on the measurement of the psychological dimensions of university students 1 year after the start of the public health emergency and restrictive measures. Furthermore, in our opinion, it was urgent to opportunely estimate the impact of massive e-learning adoption (fully online format) in academic programs for the long-term (more than 12 months).

The first interesting finding of the study was to verify that the prolonged exposure to the pandemic for undergraduates was not associated with the persistence of psychological symptoms, including anxiety, depression, and long-term stress. Our representative sample showed better emotional conditions for dealing with the pandemic stressor, showing no dysfunctional anxiety or stress. One index remained significant; the persistence of the dissociative experience of the pandemic as a traumatic event is a relevant sign to plan for future mental health strategies and actions.

Some authors highlighted that life science students were more resilient for anxiety and stress symptoms in the early stage of the pandemic because the specificity of their academic program made them more informed on public health [18,20]. After prolonged exposure to the public health emergency, our findings show that the negative emotional dimensions have been overcome and no difference can be associated with the academic programs. Our study suggests that the prolonged exposure to the public health emergency was dealt with by young university students by developing adaptive behaviors and recovering their positive mental health. The delay for academic programs, fear about their professional training, and academic performance [10,11,12,13,15,16,20] were the variables most affecting the emotional resilience of undergraduates in the first wave of COVID-19. As highlighted in Romeo’s study [29], university students experienced higher levels of anxiety and depression than the general population, suggesting the highest risk for mental health issues in the young generation. However, prolonged COVID-19 exposure was not related to the persistence of high anxiety levels for coronavirus spread, or to the fear of social isolation or contagion. This finding is interesting and points out the adaptive behaviors of university students reinforcing psychological resilience to positive emotional dimensions. Regarding dissociative signs, some evidence could relate to previous emotional fragility as well specific personality traits. Taking into account these psychological adaptation aspects of undergraduates, the continuation of studies without interruption was relevant. The early massive adoption of e-learning (full online format) in academic programs was the educational system’s response to the needs and fears of youth for their future. Our findings show the unexpected efficacy of this action as a protective factor for university students’ mental health in a public emergency. An affinity for e-learning (a positive feeling for innovative learning based on digital skills) mediated the stress and anxiety symptoms among youth involved in advanced education, and being an LS, HSS, or PES student was not significant; all students became resilient because of the positive impact of e-learning on mental health among university students. Considering all these points, we suggest the mediating e-learning influence as a ‘buffering effect’ in a public health emergency for undergraduates. The buffering effect is a process in which a psychosocial resource reduces the impact of life stress on psychological wellbeing. Having such a resource contributes to the adjustment because people are less affected by negative life events [30]. The e-learning strategy buffered the fear for the future, furthering the mental resilience of youth. According to this, an affinity for e-learning seemed to be a protective factor for the mental health of youth in the emergency. Being digitally savvy might be another relevant aspect; undergraduates are called ‘Generation Z’ and are skillful users of information and communication technology. Their widespread exposure to this technology has resulted in their comfort with and strong knowledge of digital media. Connectivity is a major factor in their social interactions, education, work, friendships, and even relationships [31,32,33].

### 4.1. Limitations

There are some limitations of the study. First, the second research protocol was based on the detection of emotional traits by online self-report measures; future research should be conducted by adopting structured measurements in traditional evaluating settings. Another limit is the unbalanced gender distribution. The reviewed studies on this topic had the participation of predominantly females; this issue should be solved by recruiting balanced gender samples.

### 4.2. Future Implications

In conclusion, the lesson learned from the pandemic is that digital innovation and investment in advanced knowledge represent two pillars for the advancement of knowledge and sciences, and more, both of them are fundamental for the health policy for the young generation.

The future challenge is the exploitation of e-learning as an innovative integrated strategy for academic programs into an empowered strategy, joining knowledge and technology.

## 5. Conclusions

In the pandemic crisis, university students, even though exposed to stressors of reduced academic relationship as well restrictive measures, have engaged their digital skills to empower their professional training, finding them to be a salient resource for their personal growth. The e-learning represented in public health emergency, the strengths for psychological resilience and emotional wellbeing of youth in academic programs. The e-learning affinity was a buffering effect for QoL and future growth.

## Figures and Tables

**Figure 1 ijerph-18-07098-f001:**
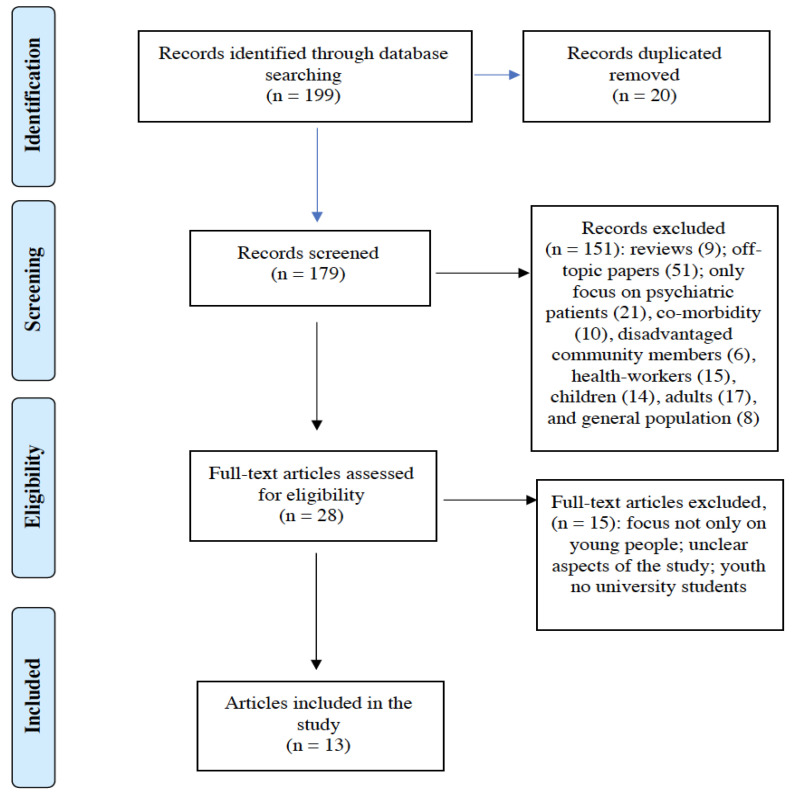
PRISMA flowchart of the study selection process for the review on the psychological impact of the COVID-19 outbreak on youth.

**Figure 2 ijerph-18-07098-f002:**
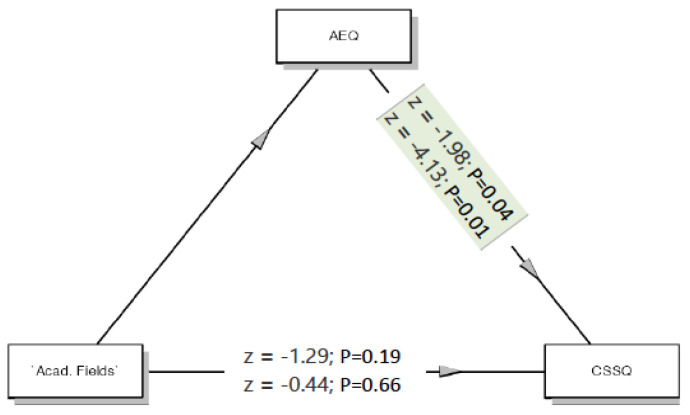
Model diagram and regression linear analysis of mediation effect of AEQ.

**Table 1 ijerph-18-07098-t001:** Main characteristics of the included studies.

Study Design	Authors	Sample Size	Sample	Recruitment
Cross-sectional study	Baloch et al. (2021) [10]	n = 494	College and university students	Online survey (WhatsApp, Email)
	Bourion-Bédès et al. (2021) [11]	n = 3928	College and university students	Online survey
	Cao et al. (2020) [12]	n = 7143	College students	Not mentioned
	Faize & Husain (2021) [13]	n = 342	University students	Online survey
	Idowu et al. (2020) [14]	n = 433	University students	Online survey
	Islam et al. (2020) [15]	n = 476	University students	Online survey (Google Forms)
	Jiang (2020) [16]	n = 472	University students	Online survey (Star software platform)
	Khan et al. (2020) [17]	n = 505	College and university students	Online survey on social media (Facebook)
	Mekonen et al. (2021) [18]	n = 350	University students	Graduating class students available during the data collection period
	Padrón et al. (2021) [19]	n = 932	University students	Online survey (internal web application)
	Sundarasen et al. (2020) [20]	n = 983	University students	Online survey
	Wan Mohd Yunus et al. (2020) [21]	n = 1005	University students	Online survey (Qualtrics survey platform)
Longitudinal study	Baiano et al. (2020) [22]	n = 25	University students	Online survey (Google Forms)

**Table 2 ijerph-18-07098-t002:** Summary of included studies evaluating the psychological impact of COVID-19.

Study Design	Authors	Measures
Cross-sectional study	Baloch et al. (2021) [10]	Zung SAS
	Bourion-Bédès et al. (2021) [11]	GAD-7, MSPSS
	Cao et al. (2020) [12]	GAD-7
	Faize & Husain (2021) [13]	GAD-7, semi-structured interview
	Idowu et al. (2020) [14]	Self-administered, semi-structured questionnaire for psychological impact of COVID 19, and their coping strategies.
	Islam et al. (2020) [15]	PHQ-9, GAD-7
	Jiang (2020) [16]	SCL-90, COVID-19 General Information Questionnaire
	Khan et al. (2020) [17]	DASS-21, IES, self-reported physical symptoms, home quarantine activities, COVID-19-related social stressors
	Mekonen et al. (2021) [18]	DASS-21
	Padrón et al. (2021) [19]	GAD-7, PHQ-9, BIT, self-perceived change in mental health
	Sundarasen et al. (2020) [20]	Zung SAS
	Wan Mohd Yunus et al. (2020) [21]	DASS-21, OHQ, WFC
Longitudinal study	Baiano et al. (2020) [22]	PSWQ, ASI-3, MAAS

Measures: Depression Anxiety Stress Scale (DASS-21), Zung Self-Rating Anxiety Scale (SAS), General Anxiety Disorder-7 (GAD-7), Patient Health Questionnaire (PHQ-9), Impact of Event Scale (IES), 90-item Symptom Checklist (SCL-90), Multidimensional Scale of Perceived Social Support (MSPSS), Oxford Happiness Questionnaire (OHQ), Work–Family Conflict Scale (WFC), Brief Irritability Test (BIT), Penn State Worry Questionnaire (PSWQ), Anxiety Sensitivity Index-3 (ASI-3), Mindful Attention Awareness Scale (MAAS).

**Table 3 ijerph-18-07098-t003:** Raw scores of psychological measurements by academic field program.

Measures	Life Sciences Mean (SD)	Physical and Engineering Sciences Mean (SD)	Human and Social Sciences Mean (SD)
PDEQ	28.0 (9.85)	28.3 (10.1)	27.1 (9.77)
CSSQ			
Global stress (Total Score)	16.6 (5.51)	16.7 (5.51)	16.1 (5.57)
Relationships and academic life	8.67 (3.63)	8.51 (3.73)	8.24 (3.84)
Isolation	5.17 (2.21)	5.34 (2.10)	5.18 (2.02)
Fear of contagion	2.72 (1.04)	2.80 (1.06)	2.68 (1.07)
CAS	5.80 (5.00)	5.99 (4.82)	6.17 (5.02)
AEQ	28.9 (9.02)	29.9 (8.75)	31.0 (9.35)

PDEQ = Peritraumatic Dissociative Experiences Questionnaire; CSSQ = COVID-19 Student Stress Questionnaire; CAS = Coronavirus Anxiety Scale; AEQ = Affinity for E-learning Questionnaire.

**Table 4 ijerph-18-07098-t004:** Post hoc comparisons (Scheffe test) academic program fields.

Comparison		SE	df	t	*p* _scheffe_	Cohen′s d
Life Science	Physical and Engineering Sciences	0.5241	2018	−1.992	0.138	−0.11
	Human and Social Sciences	0.5207	2018	−4.201	< 0.001	−0.24
Physical and Engineer Science	Human and Social Sciences	0.4631	2018	−2.469	0.048	−0.12

## Data Availability

The data presented in this study are available on request from the corresponding author. The data are not publicly available due to the exploration of the study by other ongoing researches.

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
