# Peer review of "Mediator Effect of Affinity for E-Learning on Mental Health: Buffering Strategy for the Resilience of University Students"

_ijerph, 2021, doi:10.3390/ijerph18137098_

Round 1

Reviewer 1 Report

These are two very well-conducted study protocols included in a single manuscript succinctly by the authors. They have mentioned that they conducted a narrative review for the first protocol but they have actually conducted a systematic review following the PRISMA guidelines as per my view.

In the second protocol the authors have did a wonderful job of conducting an observational study with mediation analyses. This gives a great strength to the findings. I would like the authors to add the reliability and validity information for the emotional measures used to assess the student psychological map ( lines 202 onwards). In their findings the authors mentioned that e-learning affinity for challenging and not good among life science and physics and engineering sciences due to the lab component as a part of their program and yet in the discussion based on the mediator analyses they have recommended e-learning programs as better modes of teaching and adoption for their target population. This seems a bit in conflict and needs to be clarified. Also, there needs to be a clarification of ' e-learning programs' as well. Are these fully online or hybrid or other format?

Table 4 needs a foot note of explanation.

Author Response

Dear Reviewer

firstly, we are grateful for your appreciation of our study.

Following the replying to your comments.

Many thanks 

Dina Di Giacomo

REPLYING TO COMMENTS:

REVIEWER: These are two very well-conducted study protocols included in a single manuscript succinctly by the authors. They have mentioned that they conducted a narrative review for the first protocol but they have actually conducted a systematic review following the PRISMA guidelines as per my view.

AUTHORS: We adapted to the correct form of review: from narrative review to review. Many thanks

REVIEWER: In the second protocol the authors have did a wonderful job of conducting an observational study with mediation analyses. This gives a great strength to the findings. I would like the authors to add the reliability and validity information for the emotional measures used to assess the student psychological map (lines 202 onwards).

AUTHORS: We add reliability information for emotional measures (please see pag. 7-8)

REVIEWER: In their findings the authors mentioned that e-learning affinity for challenging and not good among life science and physics and engineering sciences due to the lab component as a part of their program and yet in the discussion based on the mediator analyses they have recommended e-learning programs as better modes of teaching and adoption for their target population. This seems a bit in conflict and needs to be clarified. 

AUTHORS: It isn't in conflict. E-learning was challenging for studying performance. Our mediation analysis was oriented to the psychological effect of affinity for e-learning as strategy for personal achievement.   

REVIEWER: Also, there needs to be a clarification of ' e-learning programs' as well. Are these fully online or hybrid or other format?

AUTHORS: OK. We clarified the fully online format of e-learning in pandemic (please, see pag. 10) 

REVIEWER: Table 4 needs a foot note of explanation.

AUTHORS: Done! Please see pag. 9

Reviewer 2 Report

Please look at the attachment file.

Best regards.

Reviewer 3 Report

General comment

The present study aimed to investigate the possible negative consequences of COVID-19 outbreak on the academic quality of life of university students. In order to reach this goal, the authors first provided a review of the current literature on the mental health outcomes of COVID-19 pandemic among young adults. Secondly, they analyzed the associations between COVID-19-related risk factors and the affinity for e-learning in a large group of university students recruited from different academic field programs.

The paper deals with an interesting topic, given the negative effects that the COVID-19 outbreak could have on the mental health and academic achievement of university students. However, the manuscript currently presents some methodological weak points that should be addressed.

Specific comments

  • In the introductive section, I would suggest the authors to include further details regarding the negative impact that the COVID-19 have had on the mental health and well-being of students, in order to make the necessity of carrying out the present study clearer.
  • Moreover, in the aim description, a more appropriate definition of the first work that has been conducted should be provided. Indeed, it seems that a systematic review has been carried out, rather than a study protocol. Therefore, I would suggest the authors to clarify better this aspect. 
  • In 2.1 paragraph, I would suggest the authors to define better the type of literature review that has been conducted. Did the authors carry out a scoping review?
  • With respect to the search strategy, I think that more research terms could have been used in order to identify all the available articles that assessed the impact of the COVID-19 pandemic on University students. Indeed, some studies have been left out (e.g., Romeo, A., Benfante, A., Castelli, L., & Di Tella, M. (2021). Psychological distress among Italian university students compared to general workers during the CoViD-19 pandemic. International journal of environmental research and public health, 18(5), 2503).
  • In the statistical analysis subsection, it is not clear which types of analyses have been carried out. Did the authors employ specific software for the examination of qualitative data? In addition, why did the authors define their review as a narrative review? Indeed, it seems that a systematic literature search, using PRISMA guidelines, has been conducted.
  • In Table 2, I would suggest the authors to provide more information about the evaluation times for the longitudinal study they included. Moreover, did some of the selected articles include also a control/comparison group?
  • In 2.9 section, I would ask the authors to clarify the following sentence: “Cross-sectional studies [12-23] also explored how the COVID-19 pandemic influences on 127 the prevalence of psychological symptoms”. Indeed, it is not clear the additional results that those studies highlighted.
  • Moreover, in this section a summary of the main results the available evidence showed on the psychological effects of COVID-19 outbreak has been reported. However, besides describing those findings, also the link between this first part of the research and the following study that has been conducted should be better highlighted.
  • The authors define as a “study protocol” the second work that has been presented. However, also in this case this is not a simply presentation of a protocol, but data and results of the study have already been obtained. Therefore, the first work they carried out could be defined as a systematic (or narrative) review, while the second one as a “classic” study.
  • In the participant description, I would suggest the authors to report only how and where the participants have been recruited, moving the presentation of the main characteristics of the sample to the result section.
  • In the measure description, I would suggest the authors to report also the references for the Italian validation articles of the instruments they administered (when not present). Moreover, Cronbach’s alpha values for the present data could be provided for each of those questionnaires. Also, for the AEQ, the authors reported that a pilot study has been carried out to examine the reliability of the questionnaire. However, it is not clear where those results can be found.
  • Before reporting the results of the study, it is necessary to add a section in which the main statistical analyses performed are presented. In this way, also the following results could be easily understood.
  • In the description of ANOVA results, the F and p values for each analysis run have not been reported. Similarly, it is not clear which types of post-hoc comparisons have been performed.
  • In addition, it is not clear why a mediation model has been run. The rationale behind this analysis should be clearly explained. Also, the results of this model should be clearly detailed.
  • In the discussion section, a paragraph in which the main limitations of the present study are reported should be included. Moreover, more details could be added regarding the possible implications of the present findings for university students.
  • Please, correct some typing errors and use the term COVID-19 homogeneously throughout the text.

Author Response

My best 

Dina Di Giacomo

Reviewer 4 Report

1. line 63, did authors consider using synonyms to avoid missing papers?

2. Figure 2 could be presented better.

3. Internal consistencies were not reported for PDEQ, CSSQ, and CAS.

Author Response

Dear Reviewer

firstly, we are grateful for your appreciation of our study.

Following the replying to your comments.

Many thanks 

Dina Di Giacomo

REPLYING TO COMMENTS:

REVIEWER: 1. line 63, did authors consider using synonyms to avoid missing papers?

AUTHORS: Yes, we did! And we can confirm that our search strategy was very useful. 

REVIEWER: 2. Figure 2 could be presented better.

AUTHORS:  We copied the figure that statistical software released. If you think could be better presented, we would have your suggestions to improve our manuscript. How could be better presented the mediator analysis? 

REVIEWER: 3. Internal consistencies were not reported for PDEQ, CSSQ, and CAS.

AUTHORS: Done! Thanks

Round 2

Reviewer 2 Report

The Authors should review all previous comments, some adjustments were not finalized yet (for example the mistake in the PRISMA flow chart and in Measurements the Authors write that they investigated personality traits). I suggest to re-read previous comments and ensure that the changes are correct. In addiction I always think that the discussion is poor in content, I suggest to Author to deepen it, also expanding future implications. 

Author Response

Dear Reviewer

we appreciate definitely your support to improve the paper.

We are grateful to you. 

Below the reply to the comments and herewith attached the II revised paper

Reply to Reviewers’ Comments

Reviewer 2

The Authors should review all previous comments, some adjustments were not finalized yet (for example the mistake in the PRISMA flow chart and in Measurements the Authors write that they investigated personality traits). I suggest to re-read previous comments and ensure that the changes are correct. In addiction I always think that the discussion is poor in content, I suggest to Author to deepen it, also expanding future implications. 

Authors: We reworked the paper deeply. We dealt with all your previous feedback. Hopefully Hopeful we improved the paper as in your expectations.

Reviewer 3 Report

I really appreciate the changes that the authors have made to the paper. It has been adjusted and improved, accordingly to the reviewers’ suggestions.

I have just a few more comments for further improving the manuscript that are reported below.

  1. With regard to the search strategy, I would suggest the authors to indicate the date in which the final search has been carried out.
  2. In the measure description, I cannot find the Italian references for the questionnaires that have been administered.
  3. Thank you for the clarification about the post-hoc tests that have been used. However, the F and p values for each ANOVA run are still missing.
  4. I appreciate that the authors included a limitation paragraph in the revised version of their manuscript. However, I would suggest them not to conclude the manuscript with this subsection, but with a conclusive paragraph in which the main findings and practical implications of the present study are reported.

Author Response

Dear Reviewer

we are grateful for your support and efforts to improve the paper.

Below the replying to your comments and herewith the attached II revised paper

Reply to Reviewer Comments

Reviewer 3

  1. With regard to the search strategy, I would suggest the authors to indicate the date in which the final search has been carried out.

Authors: Done! Please see line 65

Reviewer 3

  1. In the measure description, I cannot find the Italian references for the questionnaires that have been administered.

Authors: The referenced for Italian version of measures (when available) have been inserted: please, see lines 422-425 for PDEQ. CSSQ is a measure validated on Italian population. See lines 430-433 for CAS. 

Reviewer 3

  1. Thank you for the clarification about the post-hoc tests that have been used. However, the F and p values for each ANOVA run are still missing.

Authors: Done! Please see lines 308 and 314.

Reviewer 3

  1. I appreciate that the authors included a limitation paragraph in the revised version of their manuscript. However, I would suggest them not to conclude the manuscript with this subsection, but with a conclusive paragraph in which the main findings and practical implications of the present study are reported.

Authors: We tried to work the paragraph following your suggestions. Hopeful we improved the paper as in your expectations.

Thanks!